# An Econometric Model for Measuring System-level impacts of AI on United States Power Grids

## Abstract

Data centers are a major source of power demand growth in the 21st century. Artificial Intelligence has accelerated this trend with power demand by data centers growing from 1.9% of total US power demand to 4.4% of US power in only six years. While anecdotal evidence suggests that AI data centers are using enough power to have substantial impacts on the U.S. power grids, there are no systematic studies to quantify these effects. We utilize econometric techniques to determine the impact of AI model training and inference on consumer electricity quality and fossil fuel power demand. We find significant reductions in power quality and significant increases in power demand near data centers both immediately before and immediately after the publication of AI models. The largest impact worsens power quality equivalent to an additional .5-1 power outages per year. We further show these estimates can also be used for counterfactual analysis to assess impacts of scaling for future model development.

## 1 Introduction

Over the past six years, data centers have expanded from consuming 1.9% to 4.4% of total U.S. electricity demand (Figure 1, DOE). This rapid growth—driven largely by artificial intelligence—follows two decades of flat electricity consumption and coincides with nationwide efforts to electrify transport and heating. Together, these trends are straining the power grid and raising concerns about both reliability and affordability. Recent anecdotal evidence, such as Nicoletti et al. (2024), points to data centers reducing local power quality and raising prices.

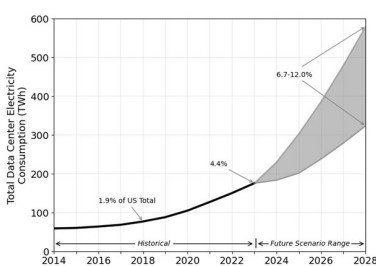

Figure 1: Projected data center growth over time. Source: DOE.

However, systematic empirical evidence on these power-grid impacts remains scarce. Existing techno-economic assessments such as those from international organizations, and industry-focused reports IEA (2024); EPRI (2024) broadly highlight increasing share of AI workloads in the data center market. At the micro-level, some studies examine the energy performance of different AI accelerators and identify workload management opportunities for reducing power consumptionShankar & Reuther (2022) and Patel et al. (2024). Last, a vast body of computer science literature focuses on the compute, energy, and environmental costs of AI model development and deployment Strubell et al. (2019); Schwartz et al. (2020); Wu et al. (2022); Berthelot et al. (2024); Morrison et al. (2025). While these provide a strong basis for understanding how AI models impact datacenter level demand and efficiency opportunities, in this work we ask questions about the impact at the power-grid level with implications for energy reliability and security.

Understanding these power grid level impacts is critical from the perspectives of energy reliability and security. In this paper, we ask seek to answer three specific questions: 1) What are the effects of

AI model releases on local power quality and power frequency? 2) How do AI training and inference affect electricity consumption around data centers owned by model developers? 3) How to analyze these grid impacts under counterfactual scenarios of efficiency and model use?

It is challenging, however, to quantify these *macro* grid-level impacts of AI models due to the lack of fine-grained data about other contributing factors that are unobservable due to proprietary aspects or otherwise prohibitively expensive to obtain. For example the power quality of a grid can be affected by many external factors including seasonal trends, location effects and data center internal factors such as other workloads which are unobservable proprietary information.

We employ econometric Difference-in-Differences (DiD) models to quantify the causal effects of AI model deployments on power systems. By comparing data centers running AI models against control groups of random data centers, and partitioning observations into pre- and post-model release periods, we isolate the impact of AI deployments while controlling for temporal and geographic factors. Using publicly available data on datacenter locations, power grids, AI model releases, weather, and market prices, we find that large AI models cause significant power quality deterioration (exceeding half the standard deviation of US power-quality distributions) and increase fossil fuel demand by terawatt hours—**equivalent to powering 100,000 homes annually**—during both training and inference phases. These estimates enable counterfactual projections for evaluating different scenarios' effects on energy consumption and grid stability.

We apply this methodology using publicly available market data on datacenter locations, power grid information, AI model release times, and other appropriate control data (e.g. weather, market prices). Specifically, we learn these DiD regressors for predicting measures of both power quality and demand. We find that large models cause significant power quality deterioration in nearby areas (well above half the standard deviation of US power-quality distributions), and increase fossil fuel demand in the order of terrawatt hours (equivalent to powering 100K homes a year) in both training and inference. We also show how these estimates can be used to evaluate different scenarios, enabling counterfactual projections of their effects on energy consumption and grid stability.

In summary, this work contributes (i) a methodology for assessing and monitoring the macro impacts of AI models on the power grid even when specific fine-grained data maybe unavailable, and (ii) the first quantitative estimates of these macro grid-level impacts of some major frontier models, adding complementary evidence to the emerging literature on the environmental and system-level impacts of AI data centers Murino et al. (2023); Guidi et al. (2024); Thangam et al. (2024).

## 2 BACKGROUND: DATACENTER CONCENTRATION AND POWER-QUALITY

The U.S. data centers are expanding rapidly, nearly tripling from its 2008 levels to 1489 active sites in 2025, with another 1,359 on the horizon Aterio (2025). The market is dominated by *hyperscalers*—vertically integrated datacenters—owned by AI heavy companies such as Amazon, Microsoft, and Google, alongside a long tail of smaller operators. Hyperscalers afford companies certain economies of scale but greatly increase the geographic concentration of power demand on the grids. Concentrated demand strains local grids in the area, raising congestion costs and sometimes degrading power quality for neighboring consumers. Large, inflexible loads from AI hyperscalers also reduce system reliability, elevate wholesale prices, and complicate renewable integration.

AI workloads affect power quality through three main mechanisms. First, data centers and AI computing facilities add substantial baseload demand that can strain grid capacity and compromise voltage regulation, especially during peak periods. Second, AI workloads fluctuate rapidly with computational needs, creating unpredictable load variations that challenge stability and frequency regulation. Third, the switching power supplies and electronics essential to AI hardware generate harmonic distortion, introducing waveform distortions that spread through distribution networks and degrade power quality. See Figure 7 in Appendix for an illustration of this harmonic distortion.

These dynamics interact with broader shifts in the electricity market: greater electrification and the transition from fossil fuels to renewables. Because renewables are intermittent and less dispatchable, rising data center loads coupled with greater renewable penetration place dual pressures on grids, shaping both market outcomes and policy debates. It is therefore crucial to quantify how much power AI uses, how it distorts quality, and the role of efficiency improvements.

## 3 METHODOLOGY

### 3.1 PROBLEM

We analyze three primary effects of artificial intelligence (AI) data centers on the power market, with a focus on how their presence and operations interact with grid stability and demand.

First, we consider the effects of AI models on the index of power quality and power frequency distortions in the vicinity of data centers. Power quality reflects the stability and reliability of the grid, particularly in terms of voltage and frequency deviations. The intensive and often irregular power draw of data centers—especially during periods of large-scale training runs—can generate fluctuations that degrade local power quality. Frequency stability is a central component of reliable electricity supply, and distortions can indicate imbalances between supply and demand. Because AI training loads are highly concentrated in time and location, they may create stress points in the grid that elevate the risk of such distortions.

To this end, we evaluate whether the deployment of new frontier models, which require extraordinary computational resources, coincides with measurable declines in regional power quality indices, thereby signaling potential challenges for grid operators. Similarly, by linking model release timelines to observed patterns in frequency deviations, we assess the extent to which the roll-out of individual models exacerbates local power quality issues beyond baseline fluctuations.

Second, we consider the effects of both training and inference workloads on overall power demand in the vicinity of data centers. Training runs, while episodic, are energy-intensive and can produce sharp spikes in consumption, whereas inference tends to generate a steadier but still substantial level of ongoing demand. Both activities alter local load profiles, raising questions about the adequacy of transmission capacity, the role of long-term contracts for renewable energy procurement, and the broader implications for regional electricity markets. By distinguishing between training and inference, we highlight how different stages of the AI lifecycle place unique and evolving pressures on power systems.

Finally, we provide analysis of the impacts of improvements in AI efficiency on power quality and power demand nearby to data centers. AI efficiency broadly defined represents improvements that can be made to reduce the energy required by models without significant corresponding decreases in quality. AI efficiency represents the main lever computer scientists have at their disposal to reduce the power consumption of AI models. We utilize our econometric estimates to determine real-world impact and combine with experimental data on the effects of hardware choices for AI to extrapolate lab estimates to meaningful impact.

### 3.2 ECONOMETRIC MODEL

Quantifying these effects is difficult given limited data on key variables. To overcome this, we draw on econometric methods, which allow credible inference even when proprietary data are unavailable.

For example, Hausman (1997) infers telecom demand from price and quantity variation, Fowlie et al. (2012) evaluate environmental regulation in electricity using emissions and price data, and Card & Krueger (1994) study minimum wage effects via cross-state employment variation. Similarly, Greenstone & Hanna (2014) analyze Indian pollution policy using ambient monitors rather than industry disclosures. These show that when markets transmit the mechanisms of interest, public data can yield the only feasible evidence. Our study follows this tradition: by exploiting observable shifts in prices, load, and capacity, we infer data center impacts that would otherwise remain opaque.

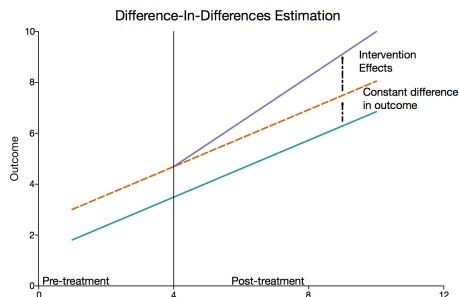

Figure 2: Difference-in-differences estimation visualized.

The particular econometric tool we use to analyze the impacts of specific AI models is *Difference-in-Differences* (DiD). Originally popularized in empirical economics through applications such as Card (2022) on training programs and Card et al. (1994) on minimum wages, difference-in-differences

has become one of the most widely used methods for causal inference with observational data. The technique estimates treatment effects by comparing the change in outcomes over time for a treated group to the change for a control group, as illustrated in Figure 2. In our application, this translates to estimating the change in generator-level power demand within a ten–square–mile radius of a data center before and after the release of a major AI model. Formally, we estimate

$$Y_{it} = \alpha + \beta \left(\text{Treatment}_i \times \text{Post}_t\right) + \gamma_i + \delta_t + \varepsilon_{it}, \tag{1}$$

where $Y_{it}$ is power demand for generator $i$ at time $t$, $\text{Treatment}_i$ indicates proximity to a data center, and $\text{Post}_t$ captures periods after the release of a major AI model. The coefficient of interest, $\beta$, measures the causal impact of model releases on local power demand. By differencing across both groups and periods, DiD controls for unobserved factors that are constant over time or shared across groups, with validity relying on the "parallel trends" assumption. The data learns the coefficient of interest by taking the difference between the average change from the initial baseline between treated and untreated regions.

### 3.3 POWER QUALITY MODEL

To analyze the effects of data centers on local power quality, we conduct two primary statistical analyses. First, we draw on Whisker Labs' measures of power quality and total harmonic distortion (THD) to estimate difference-in-differences regressions that capture the impact of AI model releases. For example, if Meta were to release a new model at a specific data center, and this activity had a causal effect on nearby power quality, we would expect to observe greater deterioration in local indices relative to areas not exposed to the release. The difference-in-differences framework allows us to test precisely this comparison by linking data on data center locations, AI model release dates, and retail electric authority service areas with Whisker Labs' measures. DiD methodology also allows us to control for sources of endogeneity. See Section 4 for more details on the data used.

Our econometric framework uses a standard difference-in-differences design to estimate the causal effects of AI model releases on grid quality. Regions with model launches serve as the **treatment group**, while those without form the **control group**. The **pre-treatment period** is the six months before a release, and the **post-treatment period** is the six months after. For demand regressions, we shorten the window to three months given the higher granularity and longer time series of the demand dataset. By contrast, power quality data are more limited, so we cannot run pre-treatment regressions, though robustness checks for alternative time intervals are provided in the appendix.

As shown in Equation 2, we model the effect on the Consumer Power Quality Index, while Equation 3 examines total harmonic distortion.

$$\text{PowerQuality}_{rt} = \alpha + \beta \left(\text{Post}_t \times \text{Treatment}_r\right) + \gamma_r + \delta_t + \varepsilon_{rt} \tag{2}$$

$$\text{TotalHarmonicDistortion}_{rt} = \alpha' + \beta' \left(\text{Post}_t \times \text{Treatment}_r\right) + \gamma'_r + \delta'_t + \varepsilon'_{rt} \tag{3}$$

The key interaction term $\text{Post}_t \times \text{Treatment}_r$ captures the difference between treatment and control regions in the post-treatment period relative to the pre-treatment period. This double-differencing approach allows us to isolate the causal effect of AI model launches by controlling for: (1) time-invariant regional characteristics through region fixed effects $\gamma_r$ (or $\gamma'_r$), and (2) common temporal trends affecting both treatment and control regions through time fixed effects $\delta_t$ (or $\delta'_t$).

The estimated coefficients $\beta$ and $\beta'$ therefore represent the average treatment effect—the causal impact of AI model releases on power quality and harmonic distortion in treatment regions during the post-treatment observation period, net of what would have occurred absent the AI launch.

### 3.4 POWER DEMAND MODEL

Power demand is a composite over fossil fuels, nuclear, and other renewables. Here we focus on fossil fuel demand partly because we have access to the necessary generator-level data and to avoid additional complexities that arise in modeling nuclear and renewable demands[1].

---

[1] It is common in economic analyses to focus on fossil fuel demand in part because renewable generation is unresponsive to demand shocks

We assess causal effects of major AI model releases on the demand for fossil fuel generation using generator-level EPA data. From here on, demand and fossil demand are used interchangeably. A key challenge in this setting is the *endogeneity problem*[2]: electricity demand and prices are jointly determined in the supply–demand system, so simple regressions may conflate the effect of AI model releases with supply-related price-driven fluctuations in generation.

To address endogeneity in our price-demand relationship, we employ **instrumental variables (IV) regression** using a **two-stage least squares (2SLS) framework**. Since supply and demand form a system with price as the common variable, simple regression cannot separate demand from supply shocks. We thus use instruments to separately identify supply and demand movements. For example, if we want to estimate how consumers respond to the price of apples, a frost that damages orchards raises prices through supply but is not directly related to consumer demand. Our instruments—generator heat rates and natural gas costs—strongly influence electricity prices through generation costs but vary independently of AI model releases, similar to how frost affects apple prices through supply but not consumer demand. These instruments' validity is established by their use in supply-side analyses in the economics literature such as Knittel et al. (2015) and Cicala (2022). In the first stage, we instrument price and AI activity variables using these exogenous factors. In the second stage, we regress fossil fuel demand on the predicted values $\widehat{\text{AITraining}}_{it}$ and $\widehat{\text{AIRelease}}_{it}$, along with instrumented price $\hat{p}$, controls $X_{it}$, generator fixed effects $\gamma_i$, and time fixed effects $\delta_t$. This approach isolates exogenous variation while controlling for weather, location, and time effects, with the price coefficient serving both as a control and for comparing dollar-equivalent effects of AI-driven demand changes.

As shown in the second-stage Equation 4, the first specification estimates the causal impact of AI training activity prior to model releases on fossil fuel demand. Another second-stage, Equation 5 then examines the post-release effects of AI activity on demand.

$$\text{FossilDemand}_{it} = \alpha + \beta \left( \text{Pre}_t \times \widehat{\text{AITraining}}_{it} \right) + \theta X_{it} + \eta \hat{p} + \gamma_i + \delta_t + \varepsilon_{it}^{\text{pre}} \qquad (4)$$

$$\text{FossilDemand}_{it} = \alpha' + \beta' \left( \text{Post}_t \times \widehat{\text{AIRelease}}_{it} \right) + \theta X_{it} + \eta \hat{p} + \gamma_i' + \delta_t' + \varepsilon_{it}^{\text{post}} \qquad (5)$$

The coefficients of interest, $\beta$ and $\beta'$, capture the average causal impact of AI activity on fossil fuel demand in the pre- and post-release periods, respectively.

### 3.5    COUNTER-FACTUAL ANALYSIS

Finally, we investigate the impacts of AI efficiency improvements on both power quality and power demand by running counterfactual analyses. The key idea is to construct regressors that capture changes in model efficiency—such as reductions in the number of parameters or computational intensity—and trace how these shifts would alter local grid outcomes. Our baseline framework already estimates how power quality responds to the release of frontier AI models; we build on this by simulating how hypothetical improvements in efficiency would change these responses. To implement this, we regress our econometric estimates on varying levels of efficiency allowing us to isolate how efficiency gains propagate into changes in electricity demand and power quality. We also extrapolate experimental data to larger models to apply estimates at the micro-scale to our estimates on impacts on power demand and quality.

### 3.6    DATACENTER ASSUMPTIONS

We test result robustness by varying treated population definitions, estimating specifications with alternative geographic boundaries to confirm consistency. This demonstrates findings reflect genuine causal impacts rather than arbitrary boundary choices. While we lack data on which models run at specific centers, our DiD methodology accounts for this by using publication dates as exogenous treatment, utilizing minimal geographic and temporal treatment windows. We plan robustness checks restricting treatment to larger data centers most likely involved in AI training. To verify treatment effects aren't noise-related, we include appendix robustness checks with randomized treatment dates as additional evidence that effects relate to model releases.

---

[2]The endogeneity problem is one where a model's errors are not random but correlated with observed or unobservable characteristics- thus making traditional estimates biased Gordon (2015)

## 4 DATA

Table 1 summarizes the datasets we use to address each of our three research questions, distinguishing between variables of interest and the control variables described above. For variables of interest, we rely on **Aterio**, which provides comprehensive information on the location and history of data centers. The Aterio dataset documents when and where centers have been established, expanded, or retired, along with details on ownership and operational capacity (in MW). This allows us to identify data centers owned by specific hyperscalers and to quantify their scale over time. We then combine these data with external sources that provide the necessary controls.

Table 1: Data sources for each analysis question.

| Source | Content | Use in Analysis | # Instances |
|---|---|---|---|
| **Q1: Difference-in-Differences (Power Quality)** | | | |
| Aterio | Data center locations, history, ownership, capacity (MW) | Define treatment regions; link AI model releases | 3862 data centers |
| Whisker Labs | CPQI (consumer reliability); THD (waveform distortion) | Outcome variables for DiD regressions | 2684 utility-months |
| EIA | Retail service territory maps | Define treatment/control boundaries | 72 utilities |
| **Q2: Instrumental Variables (Fossil Demand)** | | | |
| EPA CAMPD | Hourly generator demand; heat rates | Fossil demand outcome; heat rate as instrument | 22 million generator-hours |
| S&P Capital IQ | Natural gas prices | Instrument and fuel-cost control | 12 Million location-day prices |
| Aterio | Data center proximity to generators | Treatment near releasing-company centers | 3862 data centers |
| Meteostat | Temperature, precipitation | Demand and renewable controls | 22 million location-hours |
| ISOs | Zonal wholesale prices | Market controls | 54 zones |
| **Q3: Counterfactual Analyses (Scaling & Efficiency)** | | | |
| Whisker Labs | CPQI; THD | Baseline power quality impacts for scaling projections | |
| Epoch | Model parameters, FLOPs, release dates | Scaling regressions; efficiency counterfactuals | 75 Models |

### 4.1 VARIABLES OF INTEREST

We use three primary datasets. First, the **Consumer Power Quality Index (CPQI)** from Whisker Labs (2024) provides a composite measure of consumer-facing reliability events (surges, sags, brownouts, interruptions), summarizing frequency and severity of power deviations at the household level. Second, **Total Harmonic Distortion (THD)** data from Whisker Labs (2024) offers a *technical measure of waveform distortion*, quantifying voltage deviation from a clean 60-Hz sine wave. Elevated THD indicates grid stress, reduces motor efficiency, and shortens equipment lifespan. Figure 7 illustrates how harmonics alter voltage sine waves, reducing power reliability. Since THD data are more recent than CPQI, fewer model releases are available for THD analysis. Finally, we incorporate **generator demand** from EPA's CAMPD database, providing *hourly plant-level demand* that captures local operating characteristics. CAMPD data span 2021–2023, constraining analysis to AI model releases within that period.

### 4.2 CONTROL VARIABLES

For the **power quality regressions**, we use a parsimonious specification with only time and geographic fixed effects, which absorb seasonal patterns, long-run trends, and location-specific differences, allowing us to test whether data center activity coincides with reliability declines beyond geography and time. In contrast, the **generator demand regressions** employ a richer set of controls to address endogeneity, using *generator heat rates* from EPA CAMPD and *natural gas prices* from S&P Capital IQ Global as instruments in a two-stage least squares framework, along with CAMPD's hourly generator demand data (2021–2024). We also incorporate *Meteostat* weather variables (temperature, dewpoint, precipitation) to capture demand and renewable variability, and *ISO* market data based on boundary maps to ensure spatial consistency. Finally, *EIA retail service territory files* reconcile generator-, zonal-, and retail-level observations into a consistent framework.

### 4.3 COMBINING DATA SOURCES

To integrate datasets, we harmonize spatial and temporal units across sources. Data center locations from Aterio are mapped to ISO zones, EIA territories, and counties, enabling linkage with Whisker Labs reliability indexes (CPQI and THD) and CAMPD generator demand. CPQI has 2,683 observations (mean 0.52, s.d. 0.42), while THD is more dispersed (mean 1.81, s.d. 6.77). CAMPD generator demand averages 218 MW (s.d. 158), and wholesale prices average \$51/MWh (s.d. 148). Weather data capture local conditions (mean temperature 17°C, precipitation 0.12 mm), and time series are aligned at hourly or monthly resolution. External controls—natural gas prices, weather, and ISO market data—are merged by geography and time, yielding a unified panel suitable for difference-in-differences and IV regressions.

Our demand model focuses on deregulated markets in the Eastern Interconnection and ERCOT, where data on zones and prices are readily available, while our power quality model draws on Whisker Labs data from 72 retail utilities nationwide (monthly, 2022–2025). The demand regressions use hourly data for several thousand generators (2021–2023). We concentrate on hyperscaler AI companies—including Meta, Microsoft, Amazon, and Google—with Anthropic linked to Google due to its cloud partnership. Appendix materials include maps, sample descriptions, and data tables.

## 5 RESULTS

### 5.1 POWER QUALITY IMPACTS

Figure 3 shows the estimated impact of the release of various AI models on the local consumer power quality index (CPQI) as measured by Whisker Labs over time. The DiD coefficients (on the y-axis) represent the difference between the change in CPQI in retail electric zones with hyperscaler data centers and the change in CPQI in zones without following the release of the corresponding AI models. Higher values for the coefficients indicate deterioration in power quality that is causally attributable to the AI model releases (modulo the exhaustiveness of the controls modeled). Most coefficients are positive and significant at the 1% level with several negative but mostly insignificant coefficients.

The CPQI is an index of the expected number of surges, power outages, and brownouts weighted by the impact of the event on the home. It typically ranges from 0 to 1.2 with higher values indicating worse power quality. The most recent yearly national average CPQI was .69 with a .45 standard deviation of the power quality index distribution in U.S. **The GPT-4.5 release had an estimated power quality index impact of .327.** This level of deterioration in CPQI corresponds to moving from a typical U.S. area toward the bottom quartile of power quality—roughly **the difference between $\sim$ 1 outage/year and $\sim$ 1.5–2 outages/year for the average customer**[3]—and implies more frequent voltage anomalies that make motors and electronics run less efficiently and age faster.

Further, the continued scaling of AI models signals persistent and worsening impacts on power quality. As models grow and queries rise, these effects intensify. If the trend holds, areas near data centers may face a one standard deviation drop in power quality within a few years, implying over one day of outages annually—far above the engineering standard of one outage every ten years of Regulatory Utility Commissioners (2009). While off-peak periods in Spring and Fall may mask deterioration, regions like the Southern U.S. and PJM—already strained by summer heat—could experience outages exceeding one per month, heightening grid stress and risks of widespread failures North American Electric Reliability Corporation (NERC) (2025); Thompson (2024), as current Whisker Labs outage data suggest.

Figure 4 shows how larger AI models increase Harmonic Frequency deviations in retail electric zones with hyperscaler data centers compared to zones without. Claude 3.7 Sonnet and GPT 3.5 both worsen the Total Harmonic Distortion (THD) index by .436, pushing neighborhoods from safe power limits to dangerous levels. This increase causes home appliances like refrigerators and air conditioners to run hotter and less efficiently, potentially halving their lifespan IEE (2014); Laughner et al. (2024). The pattern indicates that larger commercial AI model deployments correlate with measurable power quality degradation, increasing consumer costs and straining the electrical grid.

---

[3]These are calculated based on Whisker Labs description of the CPQI Score with a description of derivation in the appendix.

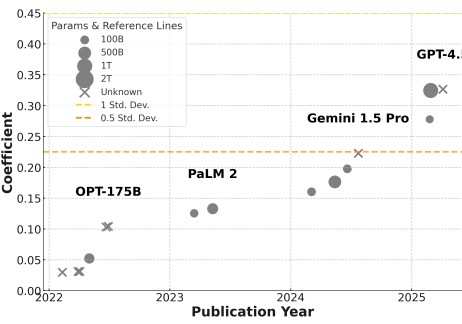

Figure 3: Impact of selected AI models on power quality over time. Plot shows substantial deterioration in power quality for recent models.

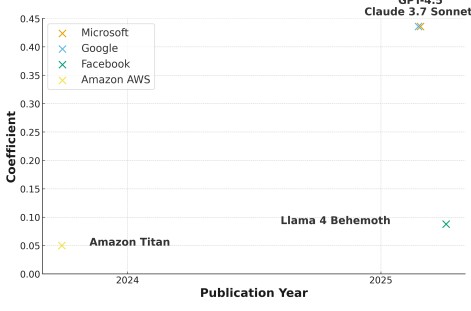

Figure 4: Impact of selected AI models on Whisker Labs' total harmonic distortions metric over Time. Plot shows increasing impact over time.

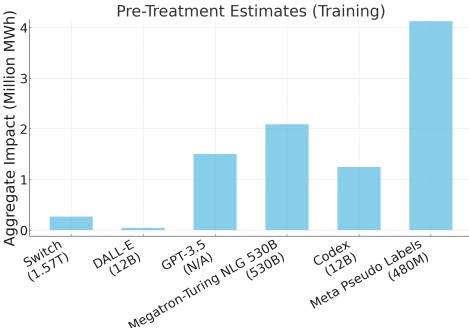

Figure 5: Impact of selected AI models on power demand for training.

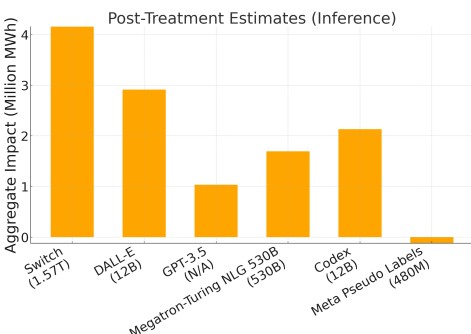

Figure 6: Impact of selected AI models on power demand for inference.

## 5.2 POWER DEMAND EFFECTS

Figures 5 and 6 visualizes our estimates of changes in power usage in MWh in the first three months before and after the release of specific large language models respectively. The pre-treatment coefficients reflect power usage (i.e. demand) of training the models, and post-treatment ones reflect that of inference.

The estimates represent the **aggregate difference between the change in the demand for generators in the ten square miles surrounding data centers compared to other areas after accounting for controls.** We see substantial increases in power demand significant at the 1% level both during training (pre-treatment), and inference (post-treatment) as shown by the large coefficient values for most models. The smaller coefficients found for some models (both positive and negative) are not statistically significant.

The largest effect on power demand is in the order of terawatt hours (i.e. millions of megawatt-hours). GPT-3.5 and DALL-E, for instance, are estimated to have increased aggregate electricity demand by **over one million and nearly three million megawatt-hours** respectively in the post-treatment period. To put this into perspective, the average U.S. household consumes about 10 to 11 megawatt-hours per year according to EIA (2024). This means the additional electricity demand attributable to GPT-3.5 in the three months post release is roughly equivalent to powering **100,000 homes for a full year**, while the impact of DALL-E corresponds to nearly **300,000 homes' annual electricity use**. These changes represent the net effects of the model release including both adoption and use of models by the public.

## 5.3 Counterfactual Analyses

Here we show an example of a counterfactual analysis that we can conduct based on the data and models we have developed. In particular, we can combine data on model size (i.e. parameter counts) and our estimates of power quality and demand to fit separate trend lines. We can then use these trend lines to extrapolate power quality impacts and demand change at other model sizes. Specific regression details in terms of model-fit and coefficient values can be found in the appendix.

To illustrate this we fit a line to power quality impacts of three models: Llama-3.1 (405 B), PaLM (540B), and Llama 4 Behemoth (2T). Using this line we see that going from a 2 Trillion model (the largest in our data set) to say a 4 Trillion model would result in a change in power quality impact from **0.321 to 0.434**, a deterioration of about 0.113 units, or just over 35 percent relative to the 2 trillion baseline. Furthermore, the relationship is exponential (the line is over log parameter counts), so a 100 billion parameter decrease has a more significant percentage-wise impact at 400 Billion parameters than at 800 Billion. The larger models become, the greater the required decrease in parameters for the same percentage decrease in power quality impact.

Using a similar process as above, we also extrapolate demand. We find that **each 1% parameter increase leads to 0.15% higher power demand** within three months of model release. Scaling from say 540 billion to one trillion parameters would **increase total power demand by 10.5%**. For DALL-E, halving parameters would **reduce inference demand by ∼300 GWh**, equivalent to the annual consumption of **27–30 thousand homes.**

Training compute requirements also have measurable effects: a one percent increase in FLOPs is associated with a **.1 percent increase in power demand**. For GPT-3.5, halving training compute FLOPs is equivalent to reducing usage by enough energy to power **8 to 10 thousand US homes for a year.** Other counterfactuals could be performed including regressing inference coefficients on reported model queries to determine the scaling effect of additional queries on model demand or determining the impacts of changing model size on carbon emissions based on the emissions mix from the data.

## 6 Conclusion

Understanding how AI models impact U.S. power grids is critical for ensuring energy reliability and security. This work introduces an econometric methodology to overcome the lack of fine-grained data about contributing factors. In particular, we use a difference-in-differences regressions to provide econometric estimates of the amount of power being used by specific AI models as well as the power distortions being caused by AI. We find evidence that AI is making power quality significantly worse and increasing fossil fuel power demand. We provide estimates of the potential impacts of increasing efficiency of AI on the power draw of data centers.

Our approach differs from the approach traditionally taken by the computer science energy efficiency literature, which tends to focus on the micro-scale to build up estimates of the costs of training a model in a specified way a specified number times. Our estimates of power demand impacts are significantly higher than previous literature estimates. Our approach also differs from the current approach amongst energy economists who tend to utilize changes from baseline simulation models to determine the impacts of external demand shocks. By focusing on market data, our approach is able to achieve estimates that are in between these levels of analysis. Our estimation is able to agnostically consider all electricity adds from AI model activities, which provides a more realistic estimate of AI life-cycle power demand. We believe this approach is also applicable for other other opaque impacts—such as the effect of large model releases on network congestion or cooling water demand—wherever direct usage data from providers is inaccessible.

While informative, this approach inherits the limitations of all Difference-in-Differences experimental designs. It relies on the parallel trends assumption, which may be violated if treated and control regions were already diverging or if generators anticipated model releases. Timing is also critical—effects may emerge gradually or with lags, complicating interpretation. Finally, contemporaneous shocks such as weather or other industrial expansions can confound results, and the treatment itself (a model release) may not map cleanly onto actual deployment. Ideally, we would like a longer power quality dataset with more granularity to make cleaner statements about long-term trends.

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

# A APPENDIX

## A.1 TOTAL HARMONIC DISTORTION

Figure 7 illustrates the distortions in the power frequencies. The green lie shows the *clear* voltage curve, which is cleanly sinusoidal, whereas with huge loads can produce harmonic distortions shown as the dotted yellow curve. This *dirtier* voltage can adversely affect the operation and lifetime of household appliances and other products that use electricity.

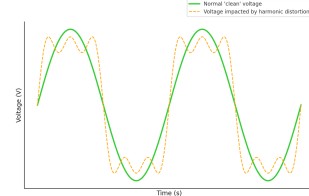

Figure 7: Total harmonic distortion representation.

## A.2 RESULTS TABLES

Table 2: Estimated Impact of AI Model Releases on Power Quality

| Model | Provider | Coefficient | Std. Error | $R^2$ |
|---|---|---|---|---|
| GPT-4.5 | Microsoft | 0.327*** | 0.087 | 0.327 |
| Llama 4 Behemoth (preview) | Facebook | 0.325*** | 0.074 | 0.313 |
| Claude 3 Opus | Google | 0.278*** | 0.043 | 0.462 |
| Gemini 1.5 Pro | Google | 0.223*** | 0.026 | 0.451 |
| GPT-4o | Microsoft | 0.198*** | 0.044 | 0.331 |
| GPT-4 | Microsoft | 0.176*** | 0.033 | 0.408 |
| Claude 3.7 Sonnet | Google | 0.161*** | 0.041 | 0.357 |
| Chinchilla | Google | 0.143*** | 0.038 | 0.570 |
| PaLM (540B) | Google | 0.142*** | 0.029 | 0.587 |
| LaMDA | Google | 0.134** | 0.044 | 0.564 |
| PaLM 2 | Google | 0.133*** | 0.024 | 0.465 |
| Claude 3.5 Sonnet | Google | 0.126*** | 0.036 | 0.378 |
| OPT-175B | Facebook | 0.104*** | 0.023 | 0.467 |
| Gemini 1.0 Ultra | Google | 0.103*** | 0.021 | 0.409 |
| Llama 3.1-405B | Facebook | 0.052 | 0.034 | 0.315 |
| Minerva (540B) | Google | 0.031 | 0.023 | 0.519 |
| Parti | Google | 0.031 | 0.023 | 0.519 |
| GPT-4 Turbo | Microsoft | 0.030* | 0.017 | 0.308 |
| GPT-3.5 | Microsoft | -0.049** | 0.018 | 0.390 |
| Flan-PaLM 540B | Google | -0.051* | 0.026 | 0.508 |
| U-PaLM (540B) | Google | -0.051* | 0.026 | 0.508 |
| Amazon Titan | Amazon AWS | -0.086*** | 0.016 | 0.380 |

Notes: Robust standard errors in parentheses. Significance levels: *** $p < 0.01$, ** $p < 0.05$, * $p < 0.1$.

Table 3: Difference-in-Differences Estimates of Data Center Announcements on Cumulative Total Queue Capacity (MW)

| | (1) No FE | (2) County & Year FE |
|---|---|---|
| Post Announcement | 714.459** | 150.741* |
| | (293.890) | (87.001) |
| Intercept | 366.406*** | -188.142*** |
| | (19.361) | (13.395) |
| Observations | 128,700 | 128,700 |
| $R^2$ | 0.010 | 0.780 |
| Adj. $R^2$ | 0.010 | 0.777 |
| Fixed Effects | No | County & Year |

Cluster-robust standard errors in parentheses.
* $p < 0.10$, ** $p < 0.05$, *** $p < 0.01$.

Table 4: Difference-in-Differences Estimates of Data Center Announcements on Cumulative Gas Queue Capacity (MW)

|  | (1) No FE | (2) County & Year FE |
|---|---|---|
| Post $\times$ Announced Capacity | 0.324*** | 0.250** |
|  | (0.105) | (0.117) |
| Intercept | 175.718*** | -43.457*** |
|  | (18.388) | (13.229) |
| Observations | 16,380 | 16,380 |
| $R^2$ | 0.012 | 0.610 |
| Adj. $R^2$ | 0.012 | 0.604 |
| Fixed Effects | No | County & Year |

Cluster-robust standard errors in parentheses.
* $p < 0.10$, ** $p < 0.05$, *** $p < 0.01$.

Table 5: Difference-in-Differences Estimates of Data Center Announcements on Cumulative Gas Queue Capacity (MW), No Interaction

|  | (1) No FE | (2) County & Year FE |
|---|---|---|
| Post Announcement | 98.976 | 273.893*** |
|  | (68.413) | (79.284) |
| Intercept | 176.560*** | 63.585*** |
|  | (18.656) | (1.28e-11) |
| Observations | 16,380 | 16,380 |
| $R^2$ | 0.003 | 0.522 |
| Adj. $R^2$ | 0.002 | 0.514 |
| Fixed Effects | No | County & Year |

Cluster-robust standard errors in parentheses.
* $p < 0.10$, ** $p < 0.05$, *** $p < 0.01$.

Table 6: Post-Treatment: Estimated Impact of AI Model Releases on Aggregate Electricity Demand

| Model | Provider | Coefficient | $p$-value | $R^2$ | Aggregate Impact (MWh) |
|---|---|---|---|---|---|
| Switch | Google | 124.46 | $5.25 \times 10^{-13}$ | 0.628 | 4,154,379 |
| DALL-E | Microsoft | 81.54 | $1.55 \times 10^{-15}$ | 0.627 | 2,910,208 |
| GPT-3.5 | Microsoft | 76.08 | $7.31 \times 10^{-6}$ | 0.622 | 1,034,517 |
| mT5-XXL | Google | 63.48 | $< 0.001$ | 0.625 | 420,630 |
| Megatron-Turing NLG 530B | Microsoft | 46.15 | $2.33 \times 10^{-4}$ | 0.628 | 1,692,209 |
| Codex | Microsoft | 35.60 | 0.0211 | 0.630 | 2,132,043 |
| Minerva (540B) | Google | 24.03 | 0.4305 | 0.624 | 1,372,621 |
| Parti | Google | 20.14 | 0.4692 | 0.625 | 1,183,787 |
| Flan-PaLM (540B) | Google | 17.36 | 0.0501 | 0.622 | 504,750 |
| U-PaLM (540B) | Google | 17.36 | 0.0501 | 0.622 | 504,750 |
| OPT-175B | Facebook | 12.85 | 0.0161 | 0.624 | 583,999 |
| ByT5-XXL | Google | 1.28 | 0.9584 | 0.624 | 66,424 |
| ProtT5-XXL | Google | -1.42 | 0.9370 | 0.629 | -64,268 |
| Meta Pseudo Labels | Google | -3.996 | 0.8284 | 0.628 | -118,203 |
| FLAN 137B | Google | -4.97 | 0.7971 | 0.624 | -167,480 |
| Chinchilla | Google | -23.54 | 0.0980 | 0.628 | -834,056 |
| PaLM (540B) | Google | -24.53 | 0.0714 | 0.628 | -873,641 |
| LaMDA | Google | -27.41 | 0.1362 | 0.626 | -643,963 |
| GLaM | Google | -27.76 | 0.3485 | 0.625 | -750,319 |
| Gopher (280B) | Google | -30.32 | 0.3113 | 0.625 | -813,756 |

Notes: Coefficients and p-values are taken from `post_coefficient` and `post_p_value`. Aggregate impact uses `post_aggregate_demand_increase`.

Table 7: Pre-Treatment: Estimated Impact of AI Model Releases on Aggregate Electricity Demand

| Model | Provider | Coefficient | $p$-value | $R^2$ | Aggregate Impact (MWh) |
|---|---|---|---|---|---|
| Switch | Google | 89.77 | $3.57 \times 10^{-4}$ | 0.628 | 268,851 |
| DALL-E | Microsoft | 66.90 | $1.13 \times 10^{-5}$ | 0.627 | 43,755 |
| GPT-3.5 | Microsoft | 39.60 | $< 0.001$ | 0.622 | 1,501,640 |
| Megatron-Turing NLG 530B | Microsoft | 35.96 | 0.0138 | 0.628 | 2,088,496 |
| Codex | Microsoft | 28.63 | 0.0108 | 0.630 | 1,245,855 |
| Minerva (540B) | Google | -22.51 | 0.0972 | 0.624 | -781,775 |
| Parti | Google | -22.94 | 0.1056 | 0.625 | -734,906 |
| Flan-PaLM (540B) | Google | 29.16 | 0.3543 | 0.622 | 1,475,972 |
| U-PaLM (540B) | Google | 29.16 | 0.3543 | 0.622 | 1,475,972 |
| OPT-175B | Facebook | -21.77 | 0.2825 | 0.624 | -502,990 |
| ByT5-XXL | Google | -6.20 | 0.7410 | 0.624 | -177,165 |
| ProtT5-XXL | Google | 101.54 | $2.26 \times 10^{-6}$ | 0.629 | 3,280,303 |
| Meta Pseudo Labels | Google | 168.29 | $1.33 \times 10^{-13}$ | 0.628 | 4,122,856 |
| FLAN 137B | Google | 3.41 | 0.8961 | 0.624 | 181,878 |
| Chinchilla | Google | -23.61 | 0.4060 | 0.628 | -583,230 |
| PaLM (540B) | Google | -22.84 | 0.4174 | 0.628 | -554,841 |
| LaMDA | Google | -27.80 | 0.3750 | 0.626 | -817,029 |
| GLaM | Google | -12.52 | 0.4856 | 0.625 | -413,053 |
| Gopher (280B) | Google | -8.12 | 0.6566 | 0.625 | -271,852 |

Notes: Coefficients and p-values are taken from `pre_coefficient` and `pre_p_value`. Aggregate impact uses `pre_aggregate_demand_increase`. Dashes indicate missing values in the spreadsheet.

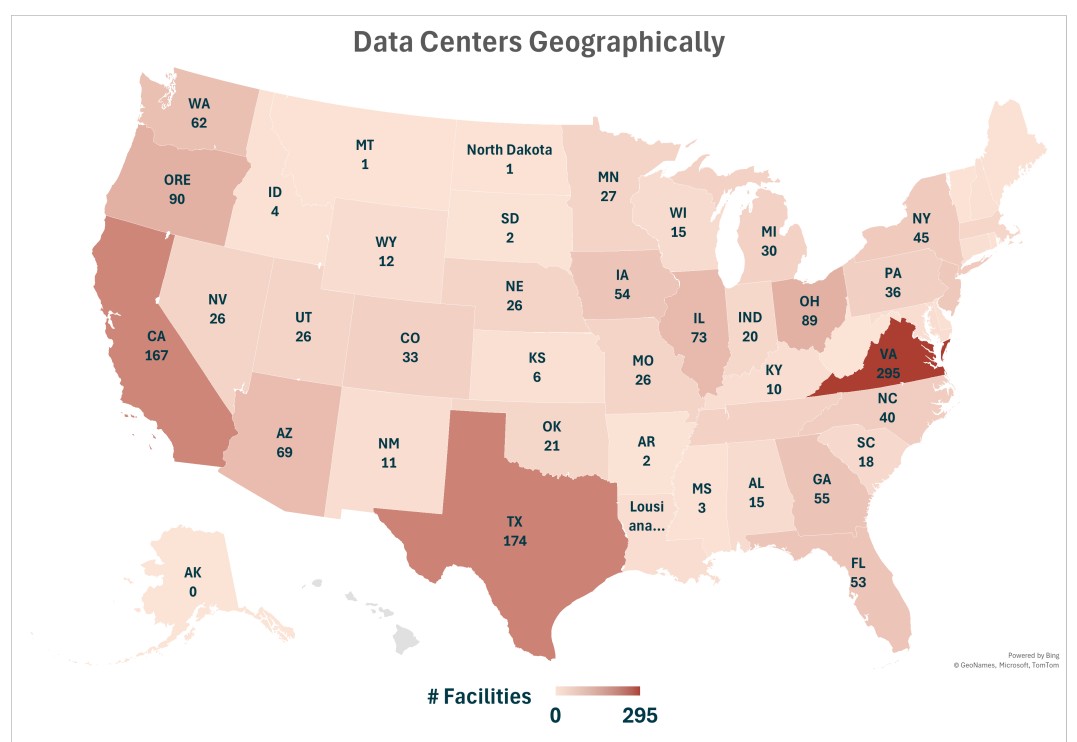

Figure 8: Data centers by state.

### A.3 EXPANDED DATASET DESCRIPTION

Below, maps can be found providing more detail on the data center dataset alongside tables on the datasets and the dataset sample selection criteria.

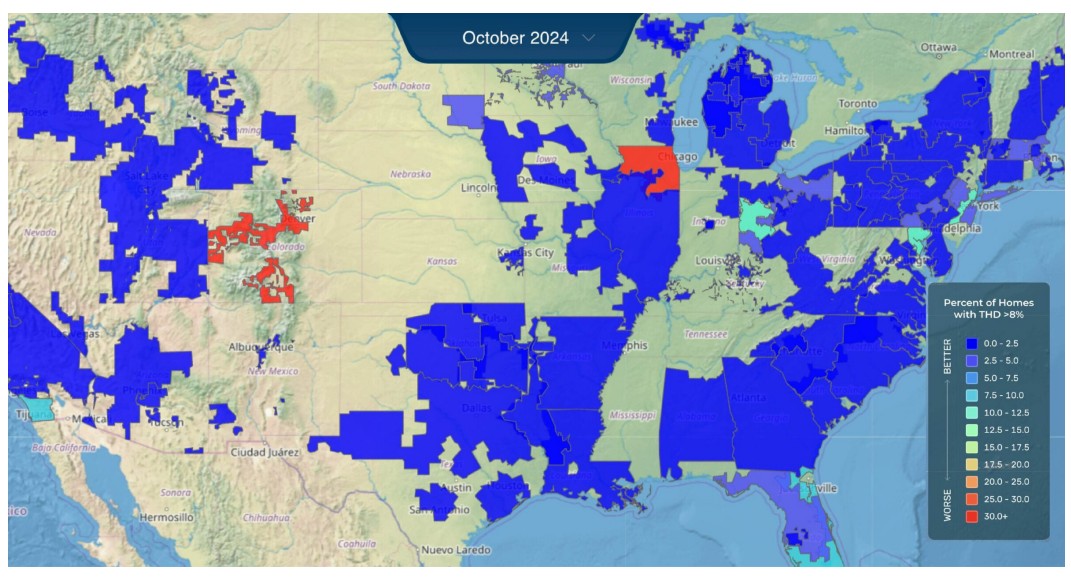

Figure 9: Map of zones for demand model.

Figure 10: Map of Retail Electric Utilities for Whisker Labs Data.

Table 8: Expanded description of data sources (for appendix).

| Source | Content and Granularity | Use in Analysis |
|---|---|---|
| **Q1: Difference-in-Differences (Power Quality)** | | |
| Aterio | Data center locations, establishment/expansion/retirement dates, ownership, and operational capacity (MW). Data are available at the facility level and updated annually. | Defines treatment regions; links AI model releases to hyperscaler-owned centers. |
| Whisker Labs | Consumer Power Quality Index (CPQI, composite measure of surges, sags, brownouts, interruptions) and Total Harmonic Distortion (THD, waveform distortion). Provided at county level, hourly frequency. | Outcome variables for DiD regressions. |
| Meteostat | Hourly temperature and precipitation, aggregated at the county level. Coverage includes all U.S. counties with weather stations. | Controls for demand fluctuations and renewable generation variability. |
| ISOs | Geographic boundary shapefiles and wholesale price data at zonal level. Boundaries digitized from ISO-provided maps; prices available hourly. | Market alignment and regional controls. |
| EIA | Retail service territory shapefiles, at utility level, updated periodically. | Used to define treatment/control boundaries and reconcile geographies. |
| **Q2: Instrumental Variables (Fossil Demand)** | | |
| EPA CAMPD | Hourly generator-level demand (MWh) and unit-specific heat rates (Btu/kWh). Coverage 2021–2023. | Generator demand is outcome; heat rates serve as instruments for prices. |
| S&P Capital IQ | Daily natural gas price series, Henry Hub benchmark. National coverage. | Instrument for price and fuel-cost control. |
| Aterio | Generator proximity to data centers owned by releasing companies, matched at 10-mile radius. | Defines treatment generators in IV design. |
| Meteostat | Hourly temperature and precipitation, as above. | Controls for demand and renewable variability. |
| ISOs | Wholesale price series, zonal level, hourly frequency. | Market-level controls. |
| **Q3: Counterfactual Analyses (Scaling & Efficiency)** | | |
| Whisker Labs | CPQI and THD (as above). Used to establish baseline deterioration in power quality. | Benchmark outcomes for scaling projections. |
| Epoch | AI model metadata: release dates, FLOPs, parameter counts. Public dataset with model-level detail. | Used for scaling regressions and efficiency counterfactuals. |
| GPU Experiments | Energy use measured on NVIDIA A6000 GPUs. Experiments run with batch size 4, sequence length 1024, and 200 inferences. Each bar in results represents total energy; lighter portion indicates communication overhead. Data collected in-house on 8xA6000 cluster. | Provides empirical GPU energy baselines for counterfactual scaling exercises. |

Table 9: Dataset sample selection and summary statistics

| Dataset | Content / Summary Statistics | Sample Selection |
|---|---|---|
| Aterio | Data center locations, ownership, and capacity; mapped to ISO zones, EIA service territories, and counties | Hyperscaler data centers (Meta, Microsoft, Amazon, Google; Anthropic via Google) |
| Whisker Labs | Reliability measures: CPQI ($N = 2{,}683$, mean=0.52, s.d.=0.42); THD (mean=1.81, s.d.=6.77) | 72 retail electric utilities, monthly data 2022–2025 |
| CAMPD | Generator-level demand: mean 218 MW (s.d.=158); operating times $\approx 1$ | Several thousand fossil generators, hourly data 2021–2023 |
| ISOs | Wholesale electricity prices: mean $51/MWh (s.d.=148); zonal boundaries | Deregulated markets in Eastern Interconnection and ERCOT |
| Weather (Meteostat) | Temperature (mean=17°C, s.d.=11.4; precipitation (mean=0.12 mm, s.d.=0.91) | Matched by county/zone, hourly or daily resolution |
| External Controls | Natural gas prices, ISO market data, geography/time harmonization | Used for both DiD and IV regressions |

Notes: Time series are harmonized at hourly or monthly resolution depending on source. Unified panel supports both difference-in-differences (power quality) and instrumental variable (demand) regressions. Maps of sample regions provided in the appendix.

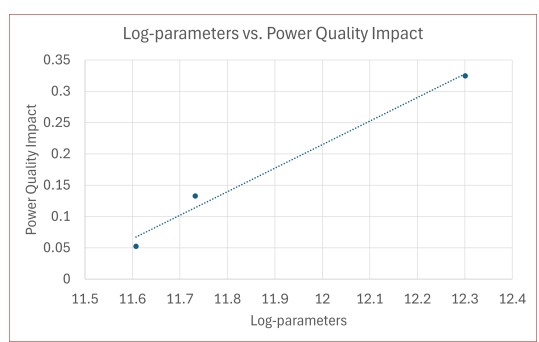

Figure 11: Power Quality Impact for Counterfactuals.

### A.4 COUNTERFACTUAL

Figure 11 shows the power quality line we fit for performing counterfactuals.

### A.5 CALCULATION OF COMPARISONS

We calculate the household average energy usage based on data provided by EIA (2024). We simply divide our estimates by the provided numbers to get the number of households for our comparisons.

### A.6 MODEL

We present a model of the impacts of varying decision choices for carbon abatement by a large hyperscaler. We present a toy model to illustrate model functioning followed by a full model. We focus on marginal carbon intensity (MCI) and average carbon intensity (ACI). MCI represents the emissions from a particular hour while average carbon intensity represents the total emissions divided by the total generation.

### A.7 TOY MODEL

**Agents and primitives.** There are two periods $t \in \{1, 2\}$ and three generator agents: a fossil unit $F$, an existing renewable $R_1$, and a potential renewable entrant $R_2$. Figure 12 showcases the agents and their interactions with the market. The hyperscaler has inelastic demand $D_t^M = 1$ in

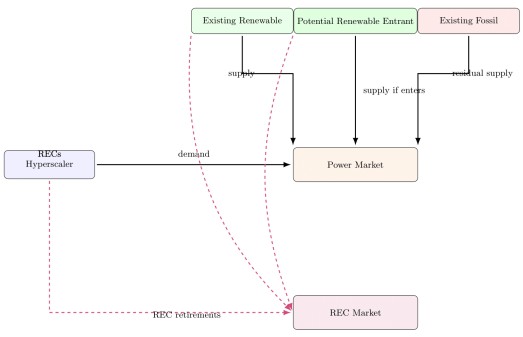

Figure 12: Agents in Model

each period, and there is no outside demand ($D_t^O = 0$). Variable costs and emissions:

$$c_F > 0, \quad e_F > 0, \qquad c_{R_1} = c_{R_2} = 0, \quad e_{R_1} = e_{R_2} = 0.$$

Capacities/output (deterministic to highlight timing):

$$x_{R_1,1} = 1, \ x_{R_1,2} = 0; \qquad x_{R_2,1} = 0, \ x_{R_2,2} \in \{0, 1\};$$

$$x_{F,t} \in \{0,1\}.$$

Renewables are *complementary in time*: $R_1$ only produces in $t = 1$, while $R_2$ (if it enters) only produces in $t = 2$. The fossil unit can meet any residual demand.

**Dispatch and prices.** Competitive dispatch follows merit order. If a renewable is available in $t$, it clears the 1 MWh load at $p_t = 0$; otherwise $F$ clears at $p_t = c_F$. Hence, without $R_2$ entry:

$$p_1 = 0, \qquad p_2 = c_F, \qquad \text{MCI}_1 = 0, \ \text{MCI}_2 = e_F.$$

With $R_2$ entered, $p_2 = 0$ as well.

**REC mechanics and the timing parameter.** Each MWh of renewable generates one REC in its production period. Let $S_t$ be REC supply, so without entry $S_1 = 1$, $S_2 = 0$; with entry $S_2 = 1$. The hyperscaler targets $100\%$ coverage ($\phi = 1$) under REC procurement with granularity parameter $h \in [0,1]$: $h = 0$ (annual matching, no timing), $h = 1$ (hourly matching). Let $R_t^G$ be granular retirements and $R^A$ annual-bucket retirements. Feasibility:

$$\underbrace{R_t^G \geq h D_t^M}_{\text{granular}}, \qquad \underbrace{R^A \geq (1-h)(D_1^M + D_2^M)}_{\text{annual}},$$

**Potential entrant $R_2$.** If $R_2$ enters, it produces $x_{R_2,2} = 1$ in $t = 2$. It pays entry cost $I > 0$ financed at $1 + r(\sigma^2)$. We compare three procurement environments (which map to different cash-flow risk for $R_2$):

1. *REC/merchant (no contract).* $R_2$ sells energy at $p_2$ and its REC at $p_2^{REC}$.
2. *PPA.* $R_2$ receives a fixed transfer $\bar{p}$ per MWh in $t = 2$ (energy+REC bundled); residual merchant exposure is zero in this toy case.
3. *Colocation.* $R_2$ is fully contracted on-site at transfer $\tilde{p}$ per MWh in $t = 2$ (no market risk).

We keep discounting trivial (two periods, take $\beta = 1$) to focus on timing and entry.

**Outcomes by procurement regime.**

## A.8 Full Model

**Agents and primitives.** We study a market with a large data center hyperscaler (e.g. Microsoft) and a set of generators indexed by owner $k$ and technology $l \in \{R, F\}$ (renewable or fossil).[4] The hyperscaler procures electricity to minimize expected total costs by choosing (i) a long-term power supply arrangement $i \in \mathcal{I}$ and (ii) an emergency backup option $j \in \mathcal{J}$.

**Key objects.**

- For each generator $g$: capacity $O_g^{\max}$, variable cost $c_g$, emissions rate $e_g$, and stochastic availability.
- For the hyperscaler: load $\{D_t^M\}_{t=1}^2$, backup option $j \in \{\text{none}, \text{diesel}, \text{storage}\}$, and procurement $i \in \{\text{REC}, \text{PPA}, \text{Colocation}\}$.

## A.9 Timing

**Stage 0a (commitment):** The hyperscaler commits to procurement $i \in \mathcal{I}$ and backup $j \in \mathcal{J}$ (contract terms public).

**Stage 0b (entry):** A set of potential generators of both types observes $(i, j)$ and decides whether to enter. Entry costs are sunk upon entry.

**Stages 1–2 (operations):** In each period $t = 1, 2$, shocks realize (demand, renewable availability). The energy market dispatches competitively; RECs are issued, banked, and retired subject to granularity rules (the "timing wedge"). Agents discount with factor $\beta \in (0, 1)$ across the two operating periods.

---

[4]Index individual plants by $g$ when convenient; technology $l(g) \in \{R, F\}$.

A.10   HYPERSCALER PROBLEM (COMMITMENT STAGE)

The hyperscaler chooses $(i, j)$ to minimize expected total costs:

$$\pi_{\text{hyper}} = \min_{i \in \mathcal{I}, \, j \in \mathcal{J}} \left[ C_i^{\text{cap}} + C_i^{\text{op}} + P_{ij} \, C^{\text{out}} + \bar{C}_{ij}^{\text{env}} \right], \tag{6}$$

where $P_{ij} = P_i P_j$ is the probability that both the contracted source fails and the backup is unavailable, $C^{\text{out}}$ is the outage loss, and $\bar{C}_{ij}^{\text{env}} = \mathbb{E}[f(\Delta E_{ij}) \, C_{ij}^{\text{env}}]$ is the expected reputational cost as a function of the emissions delta $\Delta E_{ij} \equiv E^{\text{with DC}} - E^{\text{without DC}}$.

**Contract menu.**

- *REC/merchant* ($i = \text{REC}$): the hyperscaler buys energy from the grid and retires RECs; generators remain merchant for energy and REC revenue.
- *PPA* ($i = \text{PPA}$): a renewable generator sells a fixed quantity $\bar{q}$ each period at price $\bar{p}$; residual is merchant.
- *Colocation* ($i = \text{Colo}$): a dedicated renewable unit (optionally with storage) physically colocated with the data center delivers $Q_t^{\text{colo}}$; residual met from the grid. Colocation fully insulates the generator from market risk.

**Backup.**   $j \in \{\text{none}, \text{diesel}, \text{storage}\}$ affects both reliability ($P_j$) and emissions: diesel has $e_j > 0$, storage has $e_j = 0$ if charged by colocated renewables.

A.11   ENTRY (STAGE 0B)

Potential entrants of type $l \in \{R, F\}$ decide to enter before operations. Let $I_g$ be the sunk entry cost for plant $g$, and let $r(\sigma^2)$ be the project's financing rate, strictly increasing in the variance of net operating cash flows $\sigma^2$ (a reduced-form cost-of-capital channel). For a renewable merchant entrant,

$$\Pi_g^{R,\text{merch}} = \mathbb{E}\left[ \sum_{t=1}^{2} \beta^{t-1} \big( (p_t + p_t^{REC}) x_{g,t} - c_g x_{g,t} \big) \right] - \tag{7}$$

$$I_g \left( 1 + r(\sigma_{\text{merch}}^2) \right).$$

Under a PPA for $\bar{q} \leq O_g^{\max}$ at price $\bar{p}$,

$$\Pi_g^{R,\text{ppa}} = \sum_{t=1}^{2} \beta^{t-1} \left[ \bar{p} \, \bar{q} + (p_t + p_t^{REC})(x_{g,t} - \bar{q})_+ - c_g x_{g,t} \right] - \tag{8}$$

$$I_g \left( 1 + r(\sigma_{\text{ppa}}^2) \right).$$

Under colocation (full revenue and REC certainty for the contracted amount),

$$\Pi_g^{R,\text{colo}} = \sum_{t=1}^{2} \beta^{t-1} \left[ \tilde{p}_t \, Q_t^{\text{colo}} - c_g x_{g,t} \right] - I_g \left( 1 + r(0) \right), \tag{9}$$

where $\tilde{p}_t$ is the contracted transfer for colocated output. For fossil $F$, replace $(p_t + p_t^{REC})$ by $p_t$.

**Free entry.**   With a continuum of potential entrants and competitive supply of projects, free entry implies zero-profit conditions for marginal entrants of each type/contract:

$$\Pi_g^{l, \cdot} \leq 0 \quad \text{for all } g, \qquad \Pi_g^{l, \cdot} = 0 \text{ if } g \text{ enters}, \qquad l \in \{R, F\}. \tag{10}$$

Because $r(\sigma^2)$ increases in variance and colocation eliminates variance,

$$r(0) \leq r(\sigma_{\text{ppa}}^2) \leq r(\sigma_{\text{merch}}^2), \quad \Rightarrow \quad \text{entry}_{\text{colo}} \geq \tag{11}$$

$$\text{entry}_{\text{ppa}} \geq \text{entry}_{\text{merchant}} \text{ (all else equal)}.$$

### A.12 Operations Subgame (two periods $t = 1, 2$)

**Demand.** Total load each period is $D_t = D_t^M + D_t^O$. Net grid draw by the hyperscaler is

$$G_t^M = \left(D_t^M - Q_{i,t}^{\text{contract}} - Q_t^{\text{colo}} - B_{j,t}\right)_+, \tag{12}$$

where $Q_{i,t}^{\text{contract}}$ is the (possibly firm) delivery under $i$ and $B_{j,t}$ is backup output.

**Availability.** Each generator $g$ is either fully available or off:

$$x_{g,t} \in \{0, O_g^{\max}\}, \quad \Pr(x_{g,t} = O_g^{\max}) = \alpha_{g,t}. \tag{13}$$

**Energy dispatch and price.** Given the set of available units $\mathcal{G}_t$, the competitive dispatch solves

$$\min_{\{x_{g,t}\}} \sum_{g \in \mathcal{G}_t} c_g x_{g,t} \qquad \text{s.t.} \tag{14}$$

$$\sum_{g \in \mathcal{G}_t} x_{g,t} + B_{j,t} + Q_t^{\text{colo}} = D_t, \ x_{g,t} \in \{0, O_g^{\max}\}.$$

Let $\lambda_t$ be the energy balance multiplier. The locational marginal price (LMP) is $p_t = \lambda_t = c_{g^*(t)}$, where $g^*(t)$ is the marginal unit at the optimum.

**REC issuance, banking, and the timing wedge.** Renewables mint one REC per MWh:

$$r_{g,t} = \mathbf{1}\{l(g) = R\} x_{g,t}, \qquad S_t = \sum_g r_{g,t}. \tag{15}$$

A stock of banked RECs evolves:

$$K_{t+1} = (1 - \delta)K_t + S_t - R_t^{\text{ret}}, \qquad K_t \in [0, \bar{K}], \quad t = 1, \tag{16}$$

with terminal $K_3$ free (or bounded).

To formalize matching granularity, fix a parameter $h \in [0, 1]$: $h = 0$ is annual bucket matching; $h = 1$ is full hour/period matching. Let $R_t^{M,G}$ be the hyperscaler's *granular* retirements in period $t$ and $R^{M,A}$ its *annual bucket* retirements. The hyperscaler targets a renewable share $\phi \in [0, 1]$ of its total consumption. Feasibility and obligations:

**(Granular requirement)**

$$\mathrm{R}_t^{M,G} \geq h\,\phi\,D_t^M,$$

$$\mathrm{R}_t^{M,G} \leq S_t^{\text{qual}}, \quad t = 1, 2,$$

**(Annual requirement)**

$$\mathrm{R}^{M,A} \geq (1 - h)\,\phi\,(D_1^M + D_2^M),$$

$$\mathrm{R}^{M,A} \leq K_1 + S_1 + S_2 - (R_1^{M,G} + R_2^{M,G}),$$

where $S_t^{\text{qual}} \leq S_t$ denotes RECs eligible by geography/tier. The *timing wedge* is the load not covered by same-period renewable matching:

$$w_t(h) = G_t^M - R_t^{M,G} \quad (\geq 0 \text{ if } h > 0 \text{ binds}), \tag{17}$$

$$W(h) = \sum_{t=1}^2 \mathbb{E}[w_t(h)].$$

**REC market clearing and prices.** Given banking equation 16 and obligations equation A.12–equation A.12, the competitive REC allocation minimizes present cost of compliance:

$$\min_{\{R_t^{\text{ret}}, K_2, R_t^{M,G}, R^{M,A}\}} \sum_{t=1}^{2} \beta^{t-1} p_t^{REC} R_t^{\text{ret}} + \sum_{t=1}^{2} \beta^{t-1} \overline{p}_t^{\text{ACP}} \text{short}_t \tag{18}$$

$$\text{s.t.} \quad equation\ 16,\ equation\ A.12,\ equation\ A.12,$$

$$R_t^{\text{ret}} = R_t^{M,G} + \mathbf{1}\{t=2\}R^{M,A} + R_t^{\text{LSE}} - \text{short}_t,$$
$$\text{R}^{\cdot} \geq 0,\ \text{short}_t \geq 0.$$

Let $\mu$ be the annual-constraint multiplier and $\nu_t$ the granular multipliers. Then the effective shadow value of a qualifying REC used in $t$ is

$$\tilde{p}_t^{REC} = p_t^{REC} + \nu_t + \mathbf{1}\{t=2\}\,\mu. \tag{19}$$

With interior banking, the Euler condition implies

$$p_1^{REC} = \beta(1-\delta)\,\mathbb{E}\left[p_2^{REC}\right], \tag{20}$$

and $p_t^{REC} \leq \overline{p}_t^{\text{ACP}}$ with equality if shortfalls occur.

**Emissions and carbon intensity.** Total emissions in $t$ are

$$E_t = \sum_{g \in \mathcal{G}_t} e_g x_{g,t} + e_j B_{j,t}. \tag{21}$$

Define average and marginal carbon intensity:

$$\text{ACI}_t = \frac{E_t}{\sum_g x_{g,t} + B_{j,t}}, \qquad \text{MCI}_t = e_{g^*(t)}. \tag{22}$$

Microsoft's attributable marginal emissions under $(i,j)$:

$$\text{MME}_{i,j} = \sum_{t=1}^{2} \beta^{t-1}\,\mathbb{E}\left[\text{MCI}_t \cdot G_t^M\right], \tag{23}$$

with $G_t^M$ in equation 12.

### A.12.1 EQUILIBRIUM

Given $(i,j)$ and the set of entrants from Stage 0b, a *competitive two-period operating equilibrium* is a tuple

$$\left\{\{x_{g,t}\}, \{p_t\}, \{p_t^{REC}\}, K_2, \{R_t^{M,G}\}, R^{M,A}\right\}_{t=1,2}$$

such that (i) dispatch equation 14 clears energy with $p_t = c_{g^*(t)}$; (ii) REC issuance/banking/retirements satisfy equation 16, equation A.12–equation A.12, and REC prices satisfy equation 20; and (iii) backup and colocation quantities meet equation 12. A *subgame-perfect equilibrium* of the full game consists of $(i^*, j^*)$ solving equation 6, a set of entrants satisfying free-entry conditions, and a competitive operating equilibrium in each period.

### A.12.2 MECHANISMS AND EQUILIBRIUM IMPLICATIONS

Three mechanisms drive outcomes. First, procurement choices shape operational emissions through the *timing wedge* between load and renewable generation. Colocation with storage shrinks this wedge by re-timing renewable energy into high-carbon hours; diesel backup increases emissions.

Second, contract type alters financing risk: PPAs reduce $\sigma^2$ relative to RECs, and colocation eliminates it. Lower variance reduces $r(\sigma^2)$, encouraging renewable entry.

Third, equilibrium carbon intensity reflects both the short-run operational effect and the long-run investment effect.

### A.12.3 IMPLICATIONS

Our model provides a set of empirical predictions about how hyperscalers' procurement choices affect both costs and carbon intensity. On the demand side, as the spread between the cost of procuring electricity from renewable sources and from the open market widens, hyperscalers are more likely to turn to the open market and less likely to contract renewables. At the same time, as hyperscaler demand for electricity grows, the environmental cost of their consumption rises, and the benefits of carbon-free procurement mechanisms such as PPAs and colocation increase. These benefits scale with both the overall carbon intensity of grid electricity and the size of the hyperscaler's load: the dirtier and larger the load, the greater the incentive to secure clean and reputationally valuable supply.

On the supply side, the model emphasizes that because renewable generation is intermittent, incremental data center demand tends to raise fossil generation unless there is sufficient storage to shift renewable energy across periods. The timing mismatch between hyperscaler load and renewable output—the "timing wedge"—is therefore central to understanding operational emissions outcomes. Colocation with storage can shrink this wedge by re-timing renewable production to high-carbon hours, while diesel backup increases emissions during outages.

These mechanics give rise to several testable implications. First, REC purchases on their own do not alter short-run carbon intensity, since they are purely financial transfers. Their effect arises only in the medium run, if REC demand raises REC prices and induces new renewable capacity to come online. PPAs without firming behave similarly in operational terms, but they reduce generators' revenue risk and financing costs, which increases renewable entry relative to REC-only arrangements. Colocation has a more immediate impact: behind-the-meter renewable output reduces a hyperscaler's net grid draw exactly when the colocated plant is producing, lowering local marginal emissions exposure. The effect is strongest when renewable output is correlated with hyperscaler load. Adding storage to colocation further enhances this benefit by shifting surplus renewable production to periods when demand is high and marginal carbon intensity is greatest, sharply reducing emissions attributable to the data center. By contrast, reliance on diesel backup increases operational emissions whenever it is deployed.

Finally, the model predicts an investment hierarchy driven by financing risk: colocation, by fully eliminating revenue variance, induces the largest increase in renewable entry, followed by PPAs, and then REC purchases. This creates a long-run ordering in expected carbon reductions. Moreover, because colocated renewables reduce grid purchases at the hyperscaler's node, emissions fall locally, though congestion and flow adjustments may increase carbon intensity in neighboring nodes unless total renewable capacity expands.

Together, these predictions imply that (i) REC purchases yield little immediate emissions reduction but may encourage new renewable entry; (ii) PPAs accelerate renewable deployment by reducing financing risk, though without firming they do not improve operational carbon intensity in the short run; and (iii) colocation—especially when paired with storage—directly lowers hyperscaler-attributable emissions by addressing the timing wedge between renewable generation and hyperscaler demand.

