# OpenReview forum: "An Econometric Model for Measuring System-level impacts of AI on United States Power Grids"
_ICLR.cc/2026/Conference — ICLR 2026 Conference Withdrawn Submission_

### Official Review · Reviewer_Xjib · 2025-10-31

**Soundness:** 1
**Presentation:** 1
**Contribution:** 2
**Rating:** 2
**Confidence:** 3

**Summary:**

This paper presents an econometric model to estimate the causal impact of AI data center operations on power grids using publicly available data. It employs DiD and IV regression models to measure how AI models influence power quality and demand. The study found that AI data centers cause significant power quality deterioration, providing empirical results to this important study. However, there are technical issues with the DiD rigor and methodological novelty, as well as presentation problems.

**Strengths:**

- This paper studies an underexplored issue concerning AI data centers/model releases by quantifying the macro-level energy and grid reliability impacts of AI models.
- This study integrates multiple heterogeneous datasets of data center locations, power grids, fuel price, market information, which is impressive.

**Weaknesses:**

- One of the major technical issues is the problem setup, which is a bit oversimplified and lacks proper justification. It’s impressive to integrate multiple sources of data, but there some underlying shortcomings: 1) it does not capture much physical information about the grid, as most of the studied areas are interconnected through electric power infrastructure, except Texas, and the effects may propagate; 2) the geographical coverage for the demand model may not capture the full jurisdiction of certain ISOs, how this affect the findings needs careful examination or discussion; 3) is the electricity supply to the data centers studied all coming from interconnected grids or partially from co-located power plants, which may affect the analysis.
- Despite DiD being a widely used model, the parallel trends assumption may be violated and needs more careful justification. For example, other affecting factors could include confounding regional developments, weather events, and the evolving grid conditions, such as generation mix change, battery deployments (that can balance supply and demand), because of energy transition.
- Regarding the temporal alignment, is there any more information about when model training occurs and how sensitive are the results to alternate treatment timing?
- The flow of the presentation needs improvement to explain the motivation of model choice and setup choices instead of listing what’s done. A proofread is also needed to correct typos or writing issues.
- A lot of content in the Appendix, such as game-theoretic modeling is disconnected from the main body of the work and analysis, which should be integrated better.

**Questions:**

See the weaknesses.

---

### Official Review · Reviewer_yKcM · 2025-11-01

**Soundness:** 2
**Presentation:** 3
**Contribution:** 2
**Rating:** 4
**Confidence:** 3

**Summary:**

The paper targets quantifying the macro-grid-level effects of AI data centers on the U.S. power grids, and the authors use econometric techniques, more specifically, Difference-in-Differences (DiD) models, to quantify the causal effects of AI model deployments on power systems, revealing "significant reductions in power quality and significant increases in power demand near data centers both immediately before and immediately after the publication of AI models." This study could inform infrastructure planners and policymakers in power grid systems.

**Strengths:**

1. The authors claimed there were no such empirical studies prior to this work, positioning this work to be kind of new.
2. The work presents some methodological and empirical studies on an unexplored topic, deriving some findings that could inform relevant stakeholders.

**Weaknesses:**

1. The paper may not quite fit the conference's theme of "learning representations".
2. The contributions, especially to AI researchers (who are the main contributors and audience of this conference), are unclear.
3. There is a lack of dedicated related work discussion to motivate the study approach or methodology.

**Questions:**

Why do you think the paper is a good fit for this conference? Intuitively, it may better fit conferences/journals with a social-economic focus, e.g., "Technological Forecasting and Social Change"

---

### Official Review · Reviewer_D8yK · 2025-11-02

**Soundness:** 2
**Presentation:** 2
**Contribution:** 2
**Rating:** 2
**Confidence:** 3

**Summary:**

In this work, the authors seek to demonstrate how LLM training and inference tasks have affected power demand and power quality. They do so using a combination of empirical methods, specifically by employing Difference-In-Difference and Instrumental variables. Unlike other works modeling the impacts of LLMs and data center operations on power demand and power quality, they neither rely on large scale simulation or modeling of the specific operations of the data center required to complete these tasks. They make several causal claims including impacts of data center inference on power quality (ex: GPT-4.5 caused between 1 outage to 2 outages per year for the average customer) both in terms of general indexed values and in terms of specific markers. They also find that there is a highly-significant impacts of LLM training and testing.

**Strengths:**

•	Description of DiD and 2SLS and the way they isolate desired effects.
•	Description of necessary assumptions for DiD: They did a very good job describing what needs to hold for DiD to be applied reliably. Specifically, they discussed the idea of parallel trends and explained its importance.
•	They did a good job defining power quality and using reliable metrics of power quality. While there are always other ways to measure such a broad set of qualities, I think this did a good job of identifying the key ways in which data centers would affect consumers.
•	Choice of methodologies: They filled a much-needed gap in the literature by using a methodology that does not require precise accounting of data center operations. While these estimates are also needed, small mistakes or uncertainties may lead to large errors in estimated impact. Large scale simulation would likely propagate these errors further.

**Weaknesses:**

•	While parallel trends is, by its very nature, an impossible characteristic to prove, generally pre-event trends are analyzed to show that both treated and untreated groups were on a similar trajectory prior to treatment. The authors do not validate the parallel trends assumption by demonstrating similar pre-event trends.
•	There appears to be a contradiction between the 2SLS findings and the DiD findings. How can you assume parallel trends in the DiD if you have shown that for 2SLS that before the release date there are already effects? Power quality and demand are different but, as you state with regards to harmonic frequency, are closely tied together.
•	There is a notable lack of discussion regarding the first stage of the 2SLS. It is a bit confusing how the regression is being set up. More critically, there is no evidence that the instrumental variable of choice is particularly strong.

**Questions:**

The lack of validation or justification of the assumptions underlying 2SLS and DiD cast too great a doubt on the quality of the results. Particularly, the lack of justification for why pre-release training impact wouldn’t violate the parallel trends assumption is particularly worrisome. While there may be reasons this is less problematic than it appears, this needs to be argued convincingly. Furthermore, it is not clear that the instrumental variable is sufficiently strong and weak instrumental variables are shown to lead to biased findings. Some issues that may be considered in the next version:
•	Greater discussion of the first stage of the 2SLS regression and results regarding the strength of the IV.
•	A justification for why the fact that training has an impact on power quality does not invalidate the parallel trends assumption. More broadly, why doesn’t the occurrence of high training costs for LLMs invalidate the parallel trends assumption?
•	General validation of the parallel trends assumption.

---

### Official Review · Reviewer_n3F3 · 2025-11-03

**Soundness:** 3
**Presentation:** 3
**Contribution:** 2
**Rating:** 2
**Confidence:** 4

**Summary:**

The paper presents an econometric analysis of how large AI models affect U.S. power grids. Using publicly available data on data centers, energy markets, and model release dates, the authors apply Difference-in-Differences (DiD) and Instrumental Variable (IV) regressions to estimate causal impacts of AI workloads on local power quality and fossil-fuel electricity demand.

The results show significant deterioration in power quality and multi-terawatt-hour increases in energy demand following large model releases such as GPT-4.5 and DALL-E. The paper also examines how model efficiency improvements could mitigate these effects.

**Strengths:**

+ Methodology of causal-inference design combining DiD and IV methods.
+ Good use of publicly available datasets
+ Insights into AI’s grid-level energy effects.

**Weaknesses:**

- Lacks a clear ML contribution, with no no algorithmic development, model analysis, or theoretical result.  Seems better suited for an energy-systems or sustainability venue such as ACM eEnergy or IEEE Transactions on Energy Systems.
-  The “counterfactual efficiency” discussion does not propose or test new ML efficiency techniques.
- Timing of AI model releases may not accurately represent compute-intensive periods, since training happens in advance (and a new LLM is in in training during release of a trained one), while usage might ramp up over a period of time.  eg., GPT-4 reportedly finished training around August 2022, but was released in March 2023.  So anchoring to the release date may misalign the true causal event (the energy-intensive training phase) with the observed change in power demand or grid quality.

**Questions:**

Could the authors clarify whether any ML-specific efficiency or optimization model is proposed, or if the work purely applies econometric inference?
Have the authors considered extending this framework to predictive modeling of grid demand using ML rather than econometric regressions?  This might provide methodological innovations for at-scale grid-level predictions.

---

### Note · Authors · 2025-11-12

I have read and agree with the venue's withdrawal policy on behalf of myself and my co-authors.